# The Multifaceted Functions of Autophagy in Breast Cancer Development and Treatment

**DOI:** 10.3390/cells10061447

**Published:** 2021-06-09

**Authors:** Nicolas J. Niklaus, Igor Tokarchuk, Mara Zbinden, Anna M. Schläfli, Paola Maycotte, Mario P. Tschan

**Affiliations:** 1Institute of Pathology, University of Bern, CH-3008 Bern, Switzerland; nicolas.niklaus@pathology.unibe.ch (N.J.N.); igor.tokarchuk@pathology.unibe.ch (I.T.); mara.zbinden@students.unibe.ch (M.Z.); anna.schlafli@pathology.unibe.ch (A.M.S.); 2Graduate School for Cellular and Biomedical Sciences, University of Bern, CH-3012 Bern, Switzerland; 3Centro de Investigación Biomédica de Oriente (CIBIOR), Instituto Mexicano del Seguro Social (IMSS), Puebla 74360, Mexico; paola.maycotte@imss.gob.mx

**Keywords:** autophagy, differentiation, therapy resistance, metastasis, EMT, aggressiveness, breast cancer subtypes, stem cells, tumor dormancy

## Abstract

Macroautophagy (herein referred to as autophagy) is a complex catabolic process characterized by the formation of double-membrane vesicles called autophagosomes. During this process, autophagosomes engulf and deliver their intracellular content to lysosomes, where they are degraded by hydrolytic enzymes. Thereby, autophagy provides energy and building blocks to maintain cellular homeostasis and represents a dynamic recycling mechanism. Importantly, the clearance of damaged organelles and aggregated molecules by autophagy in normal cells contributes to cancer prevention. Therefore, the dysfunction of autophagy has a major impact on the cell fate and can contribute to tumorigenesis. Breast cancer is the most common cancer in women and has the highest mortality rate among all cancers in women worldwide. Breast cancer patients often have a good short-term prognosis, but long-term survivors often experience aggressive recurrence. This phenomenon might be explained by the high heterogeneity of breast cancer tumors rendering mammary tumors difficult to target. This review focuses on the mechanisms of autophagy during breast carcinogenesis and sheds light on the role of autophagy in the traits of aggressive breast cancer cells such as migration, invasion, and therapeutic resistance.

## 1. Autophagy

Macroautophagy (hereafter referred to as autophagy) is a highly conserved catabolic control mechanism responsible for the bulk degradation of intracellular molecules and cytoplasmic organelles in a lysosome-dependent manner to maintain cellular homeostasis [1,2]. Autophagy is frequently activated as an adaptive survival response to diverse stress stimuli, e.g., starvation. Low energy upon nutrient shortage is sensed by AMP-activated protein kinase (AMPK), which upon activation blocks the key negative regulator of autophagy, the mammalian target of rapamycin (mTOR), or directly activates the initiation step of autophagy by phosphorylating unc-51-like autophagy activating kinase 1 (ULK1). In addition, low levels of amino acids within the cytoplasm inhibit Ras-related GTPase (RAG)-mediated mTOR activation and thus also activate ULK1 [3,4]. The autophagic process starts with the formation of phagophores and double membrane structures, which further elongate and close to form vesicles named autophagosomes. During elongation, autophagosomes engulf intracellular components such as protein aggregates and organelles and finally fuse with lysosomes to form autolysosomes where the content is degraded via hydrolysis. The autophagic degradation of intracellular contents yields amino acids, glucose, and free fatty acids that serve as energy substrates for macromolecule synthesis [1,2] (Figure 1).

The entire catabolic process depends on a machinery of 16–20 core autophagy (ATG)-related genes, which function in complexes at distinct steps of the autophagic pathway. During the initiation of autophagy, ULK1 dissociates from the mammalian target of rapamycin complex 1 (mTORC1), leading to ULK1 autophosphorylation and activation. Activated ULK1 recruits and phosphorylates downstream targets such as FAK Family Kinase—Interacting Protein of 200 kDa (FIP200) and ATG13 to assemble the autophagy core initiator complex also known as the ULK1 complex at the phagophore assembly site (PAS) [1,5,6] (Figure 1A). The assembly of the ULK1 complex at the PAS initiates the formation of a double membrane structure called phagophore, which is also referred to as isolation membrane (IM). Membrane sources include different organelles such as the endoplasmic reticulum (ER), the plasma membrane, the Golgi apparatus, and Golgi-derived ATG9 vesicles. Upon ULK1 signaling, the ATG2–ATG9 system is recruited to the PAS where ATG2 transports lipids from the cytoplasmic ER leaflet to the cytoplasmic leaflet of the IM, while ATG9 transports lipids from the cytoplasmic to the luminal layer of the IM [7,8,9,10]. The ULK1 complex further phosphorylates Beclin1, leading to the recruitment of class III phosphatidylinositol 3-kinase (PI3K) Vacuolar Protein Sorting 34 (VPS34) and its activating kinase Vacuolar Protein Sorting 15 (VPS15) together forming the Beclin1 complex [11,12]. VPS34 generates phosphatidylinositol 3-phosphate (PI3P), which in turn is bound by the WD-repeat protein Interacting with Phosphoinositide 1 and 2 (WIPI1 and 2) proteins that serve as adaptors for the ATG7-mediated formation of the ubiquitin-like conjugation ATG5–12-16L complex (Figure 1B). Next, the microtubule-associated light chain 3 (LC3) proteins (LC3A, LC3B, LC3C) and other ATG8 homologs, GABA Type A Receptor—Associated Protein (GABARAP, GABARAPL1, GABARAPL2) are processed by ATG4 proteases into LC3-I (GABARAP-I). LC3-I is conjugated to phosphatidylethanolamine (PE) by the ATG3-7 and the ATG5-12-16L complex generating LC3-II (GABARAP-II), which is incorporated into the phagophore membrane. This allows the elongation and closure of the phagophore that is giving rise to an autophagosome (Figure 1B,C) [5,6,7]. Of note, the ATG8 members, most notably LC3B, are crucial markers to experimentally measure the autophagic activity. However, the ATG8 homologs also have autophagy independent functions, and the ATG8 ubiquitin-like conjugation system is important but not absolutely essential for the phagophore elongation and closure [8,9,10,13]. When in close proximity, the mature autophagosome fuses with lysosomes in a Snape Receptor (SNARE)-dependent manner, leading to the degradation of the autophagosomal content by lysosomal proteases (cathepsins) via hydrolysis (Figure 1D,E) [5,6,14]. Of interest, the autophagic pathway can be inhibited by several compounds at different steps. Commonly used inhibitors are 3-Methyladenine (3-MA), ULK1- and VPS34 inhibitors that abrogate autophagy at early stages (initiation, nucleation), Chloroquine (CQ), Hydroxychloroquine (HCQ), Lys05, and Bafilomycin A1 (Baf. A1) that inhibit late stages (fusion, degradation) by alkalizing the lysosomal lumen, hindering autophagosome—lysosome fusion and inhibiting lysosomal ATPases (Figure 1A,B,D). Although autophagy inhibition at the lysosomal level potently abrogates autophagic activity, it can have off-target effects due to the inhibition of other lysosomal pathways, i.e., lysosomal exocytosis, as well as endocytic and phagocytic pathways. Similarly, blocking autophagy by targeting ATGs (i.e., ULK1 or VPS34) can have off-target effects, since these genes also have autophagy unrelated functions. Nevertheless, autophagic inhibitors are commonly used to block autophagy in experimental settings, while CQ and HCQ are compounds also used in the clinic [15,16,17,18,19,20].

Starvation-induced autophagy represents a rather unspecific bulk degradation process. However, the autophagic pathway is able to degrade intracellular components in a highly selective manner. Specific proteins, protein aggregates, lipid droplets, ribosomes, or even mitochondria are selectively captured by autophagic cargo receptors such as Sequestosome 1 (SQSTM1, aka p62) or Neighbor of BRCA1 gene 1 (NBR1). These receptors subsequently direct the cargo to the growing autophagosome [21,22]. Thus, autophagy is not only important during unfavorable environmental conditions but also regulates basal cellular homeostasis, explaining its essential role in many different cell types and processes. Hence, the ablation of autophagy affects the integrity of stem and terminally differentiated cells and leads to cellular and organismal dysfunctions during embryonic development [23,24,25].

## 2. Breast Cancer

Breast cancer is the most common and one of the most lethal cancer types among women, accounting for 24.5% of all cancer cases and 15.5% of cancer-related deaths among women in the Western world [26,27]. The short-term survival of breast cancer patients is generally high with a 5-year-survival rate of 91% in non-invasive and 86% in regional-invasive breast cancers, but it decreases to 30% in highly invasive tumors. Nevertheless, long-term survivors can experience recurrence with resistance to the initial treatments paralleled by highly invasive metastasis. This phenomenon can be ascribed to the high variability of cell types within a mammary tumor—intratumoral heterogeneity—that renders the development of efficient therapies more challenging [28].

Genomic studies have identified four different breast cancer intrinsic subtypes showing differences in their molecular expression profile and clinicopathological characteristics: luminal A, luminal B, HER2-enriched, and triple-negative subgroups [29] (Figure 2). In order to decide on treatment regimens, breast cancer patients are clinically classified according to the expression levels of the following molecular markers: estrogen receptor (ER), progesterone receptor (PR), and human epidermal growth factor receptor 2 (HER2 also known as ERBB2) as well as the proliferation marker Ki-67 [29,30,31,32]. The most common breast cancer types are luminal A and B, accounting for 40% and 30% of all breast cancer cases, respectively. The former subtype shows an ER^+^/PR^+/−^/HER2^−^ characteristic and low Ki-67 expression. Luminal A tumors have the best prognosis of all breast cancer subtypes with high survival and low recurrence rate. The luminal B subtype shows similar molecular characteristics as luminal A, but it can be HER2 positive or negative and has an intermediate to high expression of proliferation markers, causing a worse prognosis than luminal A [30,33,34,35,36]. About 15% of breast cancers belong to the HER2-enriched subtype with an ER^−^/PR^−^/HER2^+^ phenotype. HER2-enriched breast cancers also show increased proliferation marker expression and have a worse outcome compared to the luminal subtypes. Triple negative breast cancers (TNBC) are ER^−^/PR^−^/HER2^−^ with mesenchymal, basal-like, and claudin-low characteristics often carrying mutations in the tumor suppressor BReast CAncer 1 (*BRCA1*) and BReast CAncer 2 (*BRCA2*) genes, which play an important role in DNA repair. This subtype can be further subdivided into basal-like 1 and 2 (BL1 and BL2), immunomodulatory (IM), mesenchymal (M), mesenchymal stem-like (MSL), and luminal androgen receptor (LAR) subtypes. TNBC is the most undifferentiated breast cancer subtype, represents 10% of all cases, and has the worst prognosis [29,30,33,37,38] (Figure 2).

It has been suggested that the different breast cancer subtypes represent different stages of healthy mammary cell development in which the initial oncogenic insult occurred. Luminal subtypes share a genetic profile with differentiated mammary cells, while the HER2-enriched subtype represents late progenitors and TNBC are analogous to early progenitors and mammary stem cells [29,30,39,40,41] (Figure 2). Importantly, breast cancer therapies depend on the stage and subtype of the tumor and range from surgical interventions, radio- and chemotherapy, to targeted therapies that specifically inhibit molecular features such as ER and HER2 expression. These targeted therapies together with increased breast cancer awareness, due to early detection screening has led to a slight increase in the overall survival rate over the past decades despite an increased incidence in breast cancer cases [26,42,43].

## 3. Autophagy in Normal Breast Development

In normal mammary cells and tissue, autophagy plays an important part in the development and differentiation of hollow lumen structures called acini in addition to homeostasis maintenance. Mammary cells in the center of the developing acini experience apoptotic signaling due to the absence of extracellular matrix attachment (anoikis), allowing the formation of hollow lumen structures. During anoikis, the depletion of ATG genes such as *ATG5* and *ATG7* in the non-tumorigenic breast epithelial cells MCF10A resulted in increased pro-apoptotic caspase-3 signaling and reduced clonogenic viability upon reattachment. Interestingly, the depletion of autophagy in detached cells caused decreased clonogenic survival when apoptosis was blocked by the overexpression of anti-apoptotic Bcl-2 proteins. Thus, detachment-mediated anoikis induces autophagic activity that protects the cells from apoptosis even in cells that already exhibit anti-apoptotic signaling [44,45] (Figure 3). Moreover, during mammary gland postnatal development, in a lactogenic differentiation model, blocking autophagy at early stages in mammary epithelial cells (MEC) via the depletion of *ATG7* impaired the bioenergetic response occurring during MEC differentiation. This maladaptation was attributed to defects in mitophagy and paralleled by mitochondrial ROS accumulation and ROS-induced gene expression activated during lactation. Inhibition of autophagy at late degradation stages via Baf. A1 or CQ resulted in the complete termination of MEC differentiation, indicating that a careful balance between autophagy, mitophagy, and bioenergetic metabolism is necessary for MEC differentiation. Thus, balanced autophagy is essential for mammary gland function and differentiation, since increased autophagy in luminal cells would decrease the apoptosis of cells, whose death is necessary for hollow lumen formation in mammary ducts, but autophagy in lactating epithelial cells supports metabolic changes and differentiation during this process [25] (Figure 3).

## 4. Autophagy in Breast Cancer Tumorigenesis and Tumor Progression

Generally, autophagy protects normal cells from various extrinsic and intrinsic stress factors that induce DNA instability and mutations, which further lead to pre-neoplastic transformation and hyperproliferation [46]. Tumorigenic stress factors that are degraded via the autophagy pathway are protein aggregates, oncoproteins, and dysfunctional mitochondria producing ROS [47,48,49,50,51,52,53,54,55]. In addition, autophagy has immunoprotective functions, since it is involved in the control of oncogenic pathogens, regulates inflammation, and participates in immune responses that inhibit the generation and proliferation of malignant cells, e.g., by supporting antigen presentation [56,57,58]. Together, autophagy is important for the maintenance of cellular integrity and protects the cells from prolonged oncogenic insults [56].

In terms of cancer initiation, autophagy is considered tumor suppressive due to its cytoprotective role. This is evidenced by an increased autophagy-related gene signature in normal mammary glands, which is lost during breast cancer progression [59] (Figure 3). Beclin1, a component of the class III PI3K complex in the nucleation step of autophagy, was the first ATG protein whose deficiency was linked to breast cancer [60]. In agreement, increased Beclin1 activity (*Becn1* knock-in mice expressing a mutated form of Beclin1 with decreased interaction to its inhibitor Bcl-2 or use of an autophagy-inducing peptide, Tat-Beclin1) prevented HER2-positive tumor xenograft progression [61]. In addition, *Becn1*^+/+^ mice displayed decreased tumorigenesis compared with *Becn1*^+/−^ mice in a Proto-Oncogene WNT1 (WNT1)-driven breast cancer model as well as in spontaneous breast cancer formation following parity [61,62]. This is supported by the fact that monoallelic *Becn1* loss led to deregulation of the mammary cell hierarchy due to immature MEC expansion and resulted in more aggressive basal and TNBC-like cancers in WNT1-driven mammary tumorigenesis [62] (Figure 3). On the other hand, it has also been shown that Beclin1 binds to the anti-apoptotic proteins Bcl-2 and Bcl-X_L_, thereby inhibiting anti-apoptotic signaling during tumorigenesis [63,64,65]. Furthermore, autophagy is involved in the modulation of oncogene signaling, for example, by degrading of the intracellular domain of the oncogene Notch1, and *BECN1* silencing decreased colony formation and anchorage-independent growth in a TNBC cell lines [66]. Thus, autophagy or Beclin1 loss of function in normal cells are related to cell transformation, while autophagy and Beclin1 activity help maintain oncogenic signaling in transformed cells.

Of note, most of the tumor-suppressive effects of autophagy in breast cancer have been attributed to Beclin1, and *BECN1* is frequently monoallelically deleted in human breast cancer cells [60,67]. Interestingly, the tumor-suppressive Forkhead Box O (FOXO) transcription factors, which are responsible for the regulation of cellular homeostasis, stem cell maintenance, and aging, have been associated with the promotion of autophagy. Studies revealed that an increased expression of Forkhead Box O3 (FOXO3) resulted in an induction of ATG proteins involved in the initiation complex such as WIPI1, WIPI2, ULK1, as well as proteins involved in the autophagosome formation such as ATG5, ATG10, ATG14, and GABARAP. In line, the loss of *FOXO3* reduced the expression of many ATGs, leading to a reduced autophagic activity. Accordingly, the absence of FOXO3 induced mammary tumor formation, suggesting that the abrogation of FOXO3-mediated autophagy could lead to mammary carcinogenesis [68,69,70].

However, depending on the experimental context, some ATGs may support tumor-suppressive traits in malignant mammary cells. As an example, low ATG7 protein levels have been found in TNBC when compared to non-tumoral tissue. The expression of ATG7 decreased proliferation and glycolysis in TNBC cell lines, and high levels of ATG7 have been associated with a better overall survival in TNBC patients [71]. On the other hand, autophagy inhibition in RAS proto-Oncogene GTPase (RAS)-transformed ATG5 knockout breast epithelial cells reduced oncogenic transformation and anchorage-independent growth. Interestingly, similar results were obtained in the RAS mutant breast cancer cell line MDA-MB-231, pointing to an oncogenic function of ATG5/7 or the autophagic pathway in RAS-driven breast cancer cells, as it has been proposed for other cancer types [72]. Thus, the oncogenic background might be important in defining the role of autophagy. In this regard, molecular breast cancer subtypes show different sensitivities to autophagy inhibition. The TNBC subtype is the most sensitive, since autophagy promotes the activation of Signal Transducer And Activator Of Transcription 3 (STAT3), which is a common oncogene activated in TNBC [73].

In addition, deletion of *FIP200* (a.k.a. *RB1CC1*), which is important in early autophagy, resulted in autophagy defects such as an accumulation of protein aggregates and dysfunctional mitochondria, and also reduced mammary tumor formation and progression in a PyMT-induced breast cancer mouse model [74]. Accordingly, inhibiting autophagy via silencing *MAP1LC3* or *BECN1* reduced the expression of cyclin D1, integrin-β1, and phosphorylated proto-oncogene tyrosine-protein kinase (SRC), which supports the cell cycle by inducing entrance into the G1 phase, thus promoting breast cancer initiation and progression [75].

Autophagy further functions in the metastatic process by supporting cancer cell survival upon extracellular matrix detachment, which is one of the first steps in metastasis formation and cancer cell migration [45,76]. However, in more advanced stages of the metastatic process, autophagy suppresses prometastatic differentiation and metastatic outgrowth in murine breast cancer models. Attenuated autophagy in circulating tumor cells can cause an accumulation of the autophagy cargo receptor NBR1 as well as of p62/SQSTM1. These aggregated cargo receptors can serve as signaling scaffolds to activate the Mitogen Associated Protein Kinase (MAPK) pathway or the nuclear factor erythroid 2-related factor 2 (NRF2) cell survival pathway. [77,78,79]. Altogether, the above-mentioned evidence demonstrates that autophagy as a catabolic process is responsible for cell homeostasis and integrity and mostly functions as a tumor-suppressor pathway in healthy mammary cells, but depending on the context and the stage of tumor progression, autophagy and/or ATG proteins can facilitate mammary tumorigenesis or cancer progression.

## 5. The Role of Autophagy in Breast Cancer Stem Cells

In malignant breast tissue, breast cancer stem cells (BCSCs) are the drivers of tumor growth and can differentiate into several cell types similar to normal mammary stem cells. BCSCs show high self-renewal capabilities, promote tumorigenesis and distant metastasis, and are resistant to various cancer therapies [80,81]. On a molecular level, BCSCs are characterized by high levels of Cluster Of Differentiation 44 (CD44), which is a cell-surface glycoprotein involved in cell adhesion, cell–cell interactions, as well as migration. In addition, BCSCs exhibit low expression of Cluster Of Differentiation 24 (CD24), which is a surface sialoglycoprotein involved in apoptosis, and they are positive for aldehyde dehydrogenases (ALDH). ALDH catalyzes the oxidation of aldehydes and is important for stem cell maintenance and chemotherapy resistance [82,83,84,85,86].

Similar to healthy mammary cells, autophagy supports cell homeostasis and cellular integrity via its catabolic function in BCSCs. This is evidenced by the fact that ALDH1-positive stem cells isolated from MCF7 mammospheres showed increased LC3B-dependent autophagic activity compared to the bulk population. Moreover, inhibiting autophagy pharmacologically or by silencing *BECN1*, *ATG7* or *ATG4A* reduced the proliferation and pluripotency of BCSC [24,80,87,88]. Accordingly, knocking down *MAP1LC3* or *ATG12* in HER2-enriched breast cancer cells reduced the CD44^high^/CD24^low^ BCSC population [40,80,89].

In TNBCs, autophagy facilitates the secretion of interleukin 6 (IL6), which is important for the maintenance of BCSC and the activation of the oncogenic STAT3 signaling pathway. This is supported by studies in murine models where the inhibition of autophagy by silencing *Fip200* (*Rb1cc1*) inhibited Stat3 signaling, subsequently reducing tumor initiation of Aldh1^+^ or CD29^High^CD61^+^ BCSC that are associated with luminal progenitors or mesenchymal stem cells, respectively. In addition, reducing autophagic activity decreased Transforming Growth Factor Beta 2 (*Tgfβ2*) and Transforming Growth Factor Beta 3 (*Tgfβ3*) mRNA levels, thereby abrogating SMAD signaling, which is essential for CD29^High^CD61^+^ BCSCs. Thus, autophagy facilitates BCSC maintenance via the IL6/STAT3 and Epidermal Growth Factor Receptor (EGFR)/STAT3, as well as TGFΒ/SMAD pathways. Interestingly, depending on the breast cancer subtype, autophagy decreases IL6 secretion and thereby possibly reduces BCSC numbers [90,91,92,93]. This indicates that depending on the breast cancer subtype, or the BCSC or progenitor involved, autophagy can have various effects on BCSC. Inhibiting autophagy using CQ caused mitochondrial dysfunction and subsequently apoptosis in BCSC due to oxidative DNA damage and impairment of effective DNA repair. These data further point to the importance of autophagy in BCSC [94]. In line, salinomycin, a WNT inhibitor in phase I/II clinical trials, disrupts lysosomal integrity, resulting in decreased autophagic activity in BCSC that led to increased apoptosis in mammospheres [95]. In contrast, rottlerin, a natural compound extracted from the kamala tree, induces early autophagy via the AMPK and Protein Kinase B (AKT)/mTOR signaling cascade, resulting in increased apoptosis in CD44^high^/CD24^low^ BCSC isolated from primary tumors [96].

Taken together, autophagy maintains the cellular integrity and pluripotency of BCSC by regulating cell homeostasis via its catabolic function and by orchestrating signaling pathways that are important for BCSC maintenance such as IL6/STAT3 signaling. In rare cases, autophagy can also inhibit BCSC by inducing apoptosis; however, this might depend on the cellular context and the molecular background of the BCSCs.

## 6. Autophagy and Breast Cancer Dormancy

Nine out of 10 breast cancer-related deaths are caused by metastasizing relapses after the initial diagnosis and first-line treatments. This is caused by the reactivation of so-called dormant breast cancer cells [97,98]. Tumor dormancy, which is the ability of tumors to become quiescent due to unfavorable conditions, is the major cause for therapy resistance, relapse, and metastasis [99]. Disseminated tumor cells, which can be found at early stages of breast cancer, can stay dormant for decades and switch to a proliferative state upon changes in the microenvironment that trigger particular signaling pathways, such as autophagy [100,101,102,103,104]. It is suggested that autophagy is upregulated during extravasation and colonization at distant sites as a response to environmental stress conditions such as nutrient depletion [105]. This is supported by the fact that autophagy is upregulated in dormant compared to proliferating breast cancer cells [97].

Under unfavorable conditions, breast cancer cells inhibit the phosphoinositide 3-kinase (PI3K) activation via the secretion of auto- and paracrine signaling factors. This results in the inhibition of AKT and mammalian target of rapamycin (mTOR), further leading to the activation of autophagy, which puts the cell into a dormant state (Figure 3). In addition, dormant breast cancer cells show increased levels of the tumor suppressor gene Diras Family GTPases 3 (*DIRAS3*, a.k.a. *ARHI*), which is known to inhibit PI3K–AKT–mTOR signaling and activate autophagy [100,106,107]. This was nicely demonstrated in an ovarian xenograft model, where the withdrawal of ARHI led to rapid tumor growth, while the presence of ARHI inhibited tumor growth. However, autophagy inhibition via hydroxychloroquine (HCQ) re-induced tumor growth (Figure 3). These findings have been validated in breast cancer cells in vitro. Ectopic expression of ARHI induced autophagy and thereby initiated tumor dormancy during chemotherapy treatment [106,107]. Moreover, ATG4, ATG7, and the ATG8 homologs are responsible for metabolic homeostasis in dormant breast cancer cells by activating the autophagic pathway. In detail, inhibiting autophagy by using HCQ or by knocking out *ATG7* in dormant metastatic breast cancer cells reduced mitophagy, leading to the accumulation of damaged mitochondria and ROS, which decreased the viability of dormant breast cancer cells and prevented the dormancy to growth switch [97] (Figure 3).

Furthermore, transcriptomic analysis of patient-derived BCSCs showed high levels of the 6-phosphofructo-2-kinase/fructose-2,6-biphosphatase 3 (*PFKFB3*), which was paralleled by an increased metastatic potential and improved self-renewal ability of TNBC and HER2 enriched breast cancers. The PFKFB3 protein regulates the rate of glycolysis by generating fructose 2,6-bisphosphate (Fru-2,6-P2), and it directly interacts with the autophagy receptor p62, making PFKFB3 a substrate for degradation in the autophagosome. Moreover, PFKFB3 stimulates cellular processes during hypoxia, promotes cell cycle progression, and blocks apoptosis. Dormant breast cancer cells express low levels of PFKFB3 but have high autophagic activity, while metastatic breast cancer cells display high levels of PFKFB3 but rather low levels of autophagy. This suggests that low levels of autophagy lead to the stabilization of PFKFB3, resulting in the switch from a dormant to an active metastatic state. This is supported by the fact that knockdown of ATGs in dormant breast cancer cells led to the stabilization of PFKFB3 [108] (Figure 3). In summary, these findings clearly indicate that autophagy facilitates dormancy in breast cancer cells.

## 7. Autophagy and Breast Cancer Cell Dissemination

Cellular migration and invasion are important physiological mechanisms for many different biological processes such as embryonic development, wound healing, immune response, and cellular homeostasis. However, the dysregulation of cellular motility such as an increased migration and invasion can result in pathologies and is a fundamental mechanism during tumor dissemination [109,110,111]. In breast cancer, disseminating cells, known as precursors of metastases, mainly invade bone, lung, brain, as well as liver tissue and are the main cause of breast cancer-related deaths [112,113].

An important process during cell dissemination from the primary tumor is the epithelial-to-mesenchymal transition (EMT). During EMT, epithelial cells undergo molecular changes, generating a loss of cell junctions and apical–basal polarity. This leads to a mesenchymal phenotype characterized by increased migration and invasion properties as well as an enriched production of extracellular matrix [114,115]. An important regulator of EMT in breast cancer cells is the chemokine IL6, which is strongly expressed in adipocytes and therefore highly abundant in mammary tissue. IL6 facilitates EMT via Janus Kinase (JAK)/STAT and MAPK signaling pathways, leading to the molecular reprograming of cells toward a more mesenchymal phenotype by reducing the expression of E-cadherin and inducing the expression of EMT-related factors such as TWIST, SNAIL, N-cadherin, and Matrix Metalloproteinase (MMP9). The secretion of IL6 is tightly regulated by autophagy, indicating that the autophagic activity in breast cancer cells and in cells from the tumor microenvironment, consisting of adipocytes or immune cells, regulates EMT via IL6/STAT and IL6 MAPK signaling [90,91,92,116].

Adhesion of cells to the extracellular matrix (ECM) is a key step in the process of migration and invasion of metastatic tumor cells. Focal adhesions (FAs) are protein complexes that connect to the ECM and act as scaffolds during the migration of metastatic cells. FAs are composed of cytoskeletal and signaling proteins such as integrins, paxilin, vinculin, and focal adhesion kinase (FAK). Thereby, FAs also have signaling functions that regulate cellular dynamics, structure, and fate [117,118]. Interestingly, autophagosomes colocalize with FAs during disassembly in migrating cells, suggesting that autophagy is involved in the disassembly of FAs. Indeed, autophagy inhibition via silencing *ATG7* or *12* in breast cancer cells resulted in an increase in FA size that was paralleled by a decreased migration rate, indicating that autophagy reduced migration by stabilizing FA. Silencing the selective autophagy receptor *NBR1* inhibited both the assembly and disassembly of FAs, suggesting that selective autophagy is responsible for FA turnover and thereby facilitates migration and invasion. This is supported by the fact that the focal adhesion protein paxilin interacts with LC3B via its LC3 interacting region (LIR) domain and that the inhibition of autophagy by silencing *ATG5* or *7* resulted in the accumulation of paxilin paralleled by increased cell motility. Moreover, the tyrosine 40 in the LIR domain of paxilin is a target of the SRC kinase, and constitutively, active SRC strongly increased the LC3B–paxilin interaction and migration in breast cancer cells [76,119]. Likewise, the GTPase RAB7, which is important for autophagosome–lysosomal fusion, phosphorylates tyrosine 118 of paxillin, leading to increased paxilin turnover via autophagy [120].

In addition, inhibiting autophagy by silencing *MAP1LC3* or *BECN1* destabilized the integrin β1 signaling pathway inhibiting the activation of SRC and the Urokinase Plasminogen Activator Surface Receptor (uPAR)–Urokinase-Type Plasminogen Activator (uPA) system, which are important mediators of cell migration and invasion [74]. Accordingly, autophagy induced migration and invasion via facilitating the nuclear translocation of an oncogenic effector of the Hippo signaling pathway, namely Yes-Associated Protein (YAP), in TNBC but not in ER-positive breast cancer cells [121].

In contrast to the migration-supportive effect of autophagy, in highly aggressive MDA-MB-231, ectopic expression of Death effector domain containing protein (DEDD) stabilizes VPS34 and Beclin1 via physical interaction, leading to the autophagy-dependent degradation of SNAIL and TWIST and to an attenuation of the EMT process. Of note, in epithelial MCF7 breast cancer cells, DEDD led to the inactivation and degradation of VPS34, and the inhibition of autophagy. This resulted in the accumulation of SNAIL and TWIST, further promoting EMT [122]. Furthermore, downregulation of the early autophagy kinase ULK1 in breast cancer cells decreased mitophagy, leading to ROS production by dysfunctional mitochondria. This resulted in ROS-mediated induction of the inflammasome, causing the release of cytokines followed by the recruitment of osteoclasts, which led to more bone metastasis in a xenograft mouse model [123]. Inhibiting autophagy by silencing *BECN1* reduced the degradation of the intracellular domain of Notch1, leading to an increased migration of breast cancer cells [65]. Interestingly, the nuclear receptor estrogen receptor β (ERβ), which is considered a tumor suppressor in breast cancer, induces Claudin-6 (CLDN6)-mediated autophagic activity via directly transactivating *CLDN6*, thereby inhibiting the migration and invasion in breast cancer [124]. In line, autophagy inhibition via 3-MA treatment increased the migration and invasion of breast cancer cells. This could be reversed by increasing LC3 and Beclin1 expression via the administration of hydroxytyrosol and oleuropein, which are natural compounds found in olive oil [125].

Taken together, autophagy plays an important role in processes that are involved in migration such as EMT and FA turnover. Thus, autophagy is important during the cellular migration of breast cancer cells. However, if autophagy supports or inhibits migration is dependent on context and breast cancer subtype.

## 8. Autophagy in Breast Cancer Therapy

The role of autophagy in breast cancer treatment and chemoresistance is of particular importance. Of note, the current review focuses on the possible role of autophagy in modulating drug responses and not other treatment modalities. Very few reports suggest that anti-mitotic drugs—for instance, paclitaxel—decrease autophagic flux in MCF7 and SKBR3 cell lines through inhibition of VPS34 and autophagosome trafficking [126]. However, given that cancer cells utilize autophagy to ensure energy supply and adaptation to hostile conditions, autophagy is more frequently induced upon exposure to therapeutic agents in tumor cells. Importantly, the effect of autophagy on tumor cells greatly depends on the cancer entity and treatment course [127]. Autophagy activation may potentiate the cytotoxic effect of the drugs or results in chemoresistance (Figure 4).

In clinical settings, the majority of patients with ER-positive invasive breast tumors receive endocrine therapy as a first-line approach, including selective estrogen receptor modulators, for instance, tamoxifen (TAM), aromatase inhibitors, or selective ER inhibitors such as fulvestrant [128]. Despite having a good response to the initial therapy, up to 50% of responding patients will develop TAM resistance [129,130]. Similarly, 70% of patients develop resistance to trastuzumab (mAb against HER2 receptor) within a year [131]. Therefore, rather toxic compounds are being widely used when managing neglected cases. These include DNA alkylating agents (cisplatin, carboplatin), topoisomerase II inhibitors (doxorubicin, epirubicin), and anti-mitotic drugs (paclitaxel, docetaxel) [127]. Furthermore, the next-generation PI3K and Cyclin-Dependent Kinase 4/6 (CDK4/6) inhibitors have been recently suggested for metastatic cancer treatment [132,133,134].

### 8.1. Autophagy Induction

A therapeutic benefit of virtually any systemic treatment relies on apoptosis induction to eliminate malignant cells. However, frequently dysregulated apoptotic signaling enables cancer cells to overcome cytotoxic effects [135]. For this type of malignant cells, autophagy induction may be an appropriate therapeutic strategy. Essentially, autophagy potentiates drug-induced cell death in caspase-3 deficient MCF7 cells [71,136,137,138]. For instance, everolimus, an mTORC1 inhibitor developed for oral administration has shown to induce cell cycle arrest via autophagy-mediated degradation of cyclin D1 in breast cancer cells [138], while the same drug promotes autophagy in aromatase inhibitor-resistant breast cancer cells contributing to treatment resistance [139]. Based on the aberrant expression of histone deacetylases (HDAC) in rather aggressive breast cancer subtypes, the application of HDAC inhibitors is emerging as a new therapy. Vorinostat, a pan histone deacetylase inhibitor, has proven to induce LC3B-dependent autophagy via the activation of cathepsin B and decrease the survival of MDA-MB-231 cells in vitro [140].

In addition to the commonly applied endocrine, targeted, and chemotherapy approaches to breast cancer, alternative approaches have been intensively addressed due to their promising pre-clinical efficiency. Mounting evidence suggests that some bioactive natural compounds are potent autophagy enhancers. For instance, resveratrol, which is abundantly found in grapes, red wine, and peanuts [141], induces autophagy via the suppression of Wnt/β-catenin signaling. Resveratrol significantly decreased the percentage of BCSCs in MCF7 and SUM159 cell populations, resulting in smaller and reduced numbers of mammospheres [142]. Furthermore, compounds such as liensinine, dauricine, and thalidezine induced excessive autophagy and subsequently autophagic cell death via the AMPK/TSC2/mTOR signaling pathway in a panel of apoptosis-deficient cells [143,144,145]. Importantly, in *ATG7*-silenced cells, drug-mediated cytotoxicity was abrogated, indicating an autophagy-dependent mechanism of action [143]. Another approach to increase drug sensitivity refers to flavonoids and their ability to inhibit cell proliferation as well as to induce apoptosis and autophagy by downregulating Phosphatidylinositol-4,5-Bisphosphate 3-Kinase Catalytic Subunit γ (PI3Kγ), which interrupts the PI3K/AKT/mTOR/ Ribosomal Protein S6 Kinase (p70S6K)/ULK pathway in breast cancer cells [146] (Figure 4).

In mammalian cells, and particularly in cancer cells, the mechanisms by which autophagy can kill cells is context and cell type dependent, since evidence does not point to a unifying cell death pathway [147]. Perhaps the best characterized autophagic cell death pathway in mammalian cells is the one termed “autosis”, in which autophagic cell death was induced by an autophagy-inducing peptide derived from Beclin1 in a subpopulation of nutrient-starved cancer cells or in neurons subjected to hypoxia–ischemia in vivo. This type of cell death was prevented by the inhibition of autophagy and was regulated by Na+,K+-ATPase [148]. The relevance of this pathway to cancer cells or to cancer treatment remains undetermined. Another remaining question is still whether the cells die “by or with” autophagy [149,150]. In addition, the concepts of caspase-independent “autophagic cell death” and “autophagy-mediated cell death” have been neither universally described nor unanimously accepted and therefore warrant further study [151,152,153].

### 8.2. Autophagy Inhibition

Autophagy activation upon anti-cancer therapy may result in cancer cell protection contributing to drug resistance and breast cancer progression via PI3K/AKT/mTOR, ERK, p53, Vascular Endothelial Growth Factor (VEGF), EGFR, and MAPK14/p38*α* signaling [154,155,156,157]. Essentially, the PI3K/AKT/mTOR pathway is likely one of the most frequently mutated pathways in human cancers; it is closely related to autophagy and therefore represents a potential point of attack [152,158]. Accordingly, the inhibition of ETS transcription Factor ELK3 (ELK3), which is frequently overexpressed in aggressive breast cancers [159], led to PI3K/AKT/mTOR activation, which in turn enhanced the chemosensitivity of MDA-MB-231 cells to doxorubicin by inhibiting protective autophagy [157].

Despite a close interplay of the involved pathways, there are increasing numbers of reports indicating that tamoxifen [160], epirubicin [161], doxorubicin [157,162], paclitaxel [163], and even the novel CDK4/6 inhibitor palbociclib [134,164] induced autophagic flux in parental and drug-resistant ER+ and TNBC cells. Knocking down key autophagy genes (*ATG5*, *BECN1*, or *ATG7*) in combination with TAM treatment resulted in decreased cell viability in MCF7 and T47D breast cancer cell lines. [160]. Depleting *ATG5* or *BECN1* in the same cells had no effect on cell viability upon palbociclib treatment, but it led to irreversible growth inhibition, G1 arrest, and enforced senescence [164].

Significantly increased autophagic activity was observed in trastuzumab-resistant SKBR3 and JIMT1 breast cancer cells bearing HER2 amplification, where it correlated with the downregulation of AKT and Extracellular Signal-Regulated Kinase 1 (ERK1)/ Extracellular Signal-Regulated Kinase 2 (ERK2) signaling during early treatment [165]. Notably, autophagy inhibition with 3-MA reduced the viability, and knockdown of LC3 decreased the proliferation of these trastuzumab-resistant cells [166]. A later study found elevated ATG12 protein levels in a cohort of trastuzumab-resistant HER2+ cells [167]. Following *ATG12* silencing, JIMT1 xenografts exhibited significantly reduced tumor growth. Moreover, the combination of trastuzumab with the autophagy/lysosome inhibitor chloroquine increased apoptosis and decreased cell viability and colony formation of JIMT1 cells in vitro and in vivo [168]. In addition, HCQ suppressed cell proliferation and migration, especially in the HER2+ SUM-190 basal breast cancer cells [77,169].

Importantly, there is a close relationship between Beclin1 and HER2 in breast cancer biology. HER2 is known to inhibit autophagy by binding to Beclin1 [61], and HER2 amplification in breast cancer is often associated with *BECN1* DNA copy loss [170]. Subsequent studies showed that Beclin1 exerts either a cytoprotective or cytotoxic effect depending on the molecular subtype [171]. For instance, the suppression of Beclin1 increased TAM sensitivity in ER-positive breast cancer cells in vitro, and lower Beclin1 expression was associated with favorable outcome in patients with ER-positive breast cancer [172]. Similarly, Beclin1 inhibition enhanced paclitaxel-mediated cytotoxicity in BT-474 ER+ breast cancer cells in vitro and contributed to decreased tumor volume in a xenograft model [171], suggesting that Beclin1 protects these cells from drug-induced apoptosis. However, it is not excluded that this effect is caused by autophagy-independent functions of Beclin1. In contrast, the overexpression of Beclin1 in highly resistant BT-549 and MDA-MB-231 breast cancer cells activated autophagy and inhibited cell proliferation during normal culture conditions and, paradoxically, increased cell survival under starvation, hypoxia, or doxorubicin stimulation [173].

Autophagy has also been related to the maintenance of cancer cell signaling related to chemoresistance or pro-tumorigenic features. The translocation of high-mobility group box 1 (HMGB1) from the nucleus to the cytoplasm facilitates the assembly of Beclin1–PI3K–III complexes via MAPK/ERK signaling, thereby mediating autophagosome formation [174,175]. HMGB1-mediated autophagy leads to increased chemoresistance in breast, ovarian, liver, and lung carcinomas in vitro. The subsequent depletion of HMGB1 or limiting HMGB1 cytosolic translocation diminishes autophagic protection in response to anti-mitotic drugs [176,177,178,179]. In line with these findings, a recent study revealed that inhibition of autophagy reduced leptin-induced cell proliferation, migration, and attenuated ERK phosphorylation [180]. In an attempt to target autophagy-mediated chemoresistance and pro-tumorigenic cancer cell signaling, encouraging in vitro results of epigallocatechin-3-gallate (EGCG), a bioactive substance derived from green tea, showed the potential to overcome drug resistance by inhibiting therapy-induced autophagy upregulation [181]. EGCG attenuated the cell viability of MDA-MB-231 cells in a dose-dependent manner by downregulating the expression of β-catenin, Cyclin D1, and phosphorylated AKT [182] (Figure 4).

Collectively, autophagy inhibition leads to significant drug re-sensitization in different settings, indicating that autophagy acts as a survival mechanism to overcome the cytotoxic effect of chemotherapy or targeted cancer drugs, thereby supporting the translational use of autophagy-modulating compounds.

## 9. Combination Therapies in the Clinic

The anti-malarial drug CQ and its derivative HCQ disrupt lysosomal functions and therefore can, among other cellular functions [183,184], block autophagy. These two compounds are the only autophagy inhibitors authorized for their use in in clinical trials. [169,184]. In breast cancer trials, treatment with CQ plus radiation therapy resulted in a one-year brain metastasis progression-free survival in 83.9% of patients compared to 53.1% in the placebo group [185]. A phase II clinical trial of HCQ monotherapy in patients with metastatic pancreatic cancer revealed that although HCQ was not potent enough to provide therapeutic benefit, a high-dose regimen was well tolerated [186]. Later, the results of a phase I local trial in a small cohort of solid cancers (nine patients) demonstrated that resected tumors from HCQ pre-treated patients showed increased p62 levels, indicating autophagy inhibition [187]. Other trials in solid cancers showed synergistic effects of HCQ with additional therapeutic agents, explaining its limited efficacy as a single agent and recommending HCQ to be applied in combination with other drugs to enhance their efficacy [77,130,188]. For instance, the combination of HCQ + Abemaciclib (CDK4/6 inhibitor) +/− endocrine therapy suggests a positive outcome for clinical trial in HR-positive/HER2- breast cancer lesions (NCT04316169) [189]. Another autophagy inhibitor Lys05 showed 10-fold higher activity than CQ in vitro and holds great promise in pre-clinical breast cancer animal models [190]. In general, Baf. A1 shows a high toxicity and can therefore not be used in the clinic [191,192].

Recently, a new class of specific CDK4/6 inhibitors has been approved for the treatment of advanced ER-positive breast cancers in combination with aromatase or selective ER inhibitors [134]. CDK4/6 inhibitors in combination with the anti-hormonal agent and HCQ or Lys05 (another chloroquine-like lysosomal inhibitor) induced senescence and resulted in significant tumor volume reduction in xenograft experiments [164]. Nevertheless, BCSCs, as a main source of cancer relapse, remained largely intact in vitro upon CDK4/6i, HR antagonists, and autophagy inhibition in ER+ breast cancer cells [164]. BCSCs persistently remained in the quiescent G0 phase unlike other cancer cells that rapidly divide. Therefore, the standard therapies, exclusively targeting replicating cells, seem to be ineffective against BCSCs [193].

Regarding autophagy induction for breast cancer therapy, a promising approach is to target the formation of Bcl-2/Bcl-X_L_-Beclin1 complexes by using selective Bcl-2 inhibitors [194] (Figure 4). For instance, Venetoclax (VCX) induces autophagy through its binding to the BH3 domain of Beclin1, competitively disrupting the interaction between Beclin1 and Bcl-2/Bcl-X_L_ [195,196]. As a proof of concept, VCX-mediated inhibitory effects on cellular survival have been recently shown in several breast cancer cell lines. In addition to inducing apoptosis and cell cycle arrest, VCX treatment increased ROS formation and ATG protein levels. Although this study does not clearly implicate autophagy in apoptosis induction, VCX-induced cellular changes sensitized MDA-MB-231 triple negative cells to doxorubicin, supporting the use of this compound for chemosensitization [197,198]. These and other [199,200] findings initiated the clinical trial combining VCX with standard-of-care TAM in patients with ER-positive and Bcl-2-positive metastatic breast cancer. The early results of a phase Ib study showed promising synergism with a tolerable safety profile, including significant clinical response in four patients who had been previously treated with CDK4/6 inhibitors [201]. Given the small size of the study (33 patients), the global, randomized, phase 2, multicenter, open-label study VERONICA (NCT03584009) has been initiated, and patient enrollment is still ongoing to test the combination of VCX with selective ER inhibitor fulvestrant [202]. Currently, there are no active clinical trials exploring the induction of cytotoxic autophagy for the treatment of breast cancer. However, several clinical trials explore the inhibition of the PI3K/AKT/mTOR pathway, which is commonly altered in TNBC or in resistant ER+ or HER2+ breast cancer subtypes [203,204,205]. Although the effects of PI3K/mTOR inhibition could be autophagy independent, it will be important to understand the contribution of autophagy to patient outcome, since mTOR inhibitors are potent activators of autophagy.

However, reluctantly, clinical testing of autophagy modulators for cancer patients is currently being evaluated in small clinical trials. A major concern is that the high concentrations of compounds required to block autophagy in cell culture may be difficult to determine and maintain in a patient setting [169]. Moreover, CQ and HCQ have been associated with irreversible visual loss due to retinal toxicity [206]. Therefore, a new generation of specific autophagy inhibitors bearing greater in vivo activity are urgently needed to conduct randomized studies, particularly in breast cancer.

## 10. Conclusions and Perspectives

In this review, we emphasized the current role of autophagy in different aspects of breast cancer pathology and treatment (Figure 5). The role of autophagy in breast cancer is versatile and depends on the biology of the breast cancer subtype. For example, Beclin1-dependent induction of autophagy inhibited tumorigenesis and proliferation in luminal-like breast cancer cells, while highly aggressive TNBC that are often driven by strong oncogenic signaling pathways require autophagy for anchorage-dependent and independent growth [60,72,74]. Nevertheless, even within the different breast cancer subtypes, autophagy may have contradictory or context-dependent roles since it also reduces tumor initiation and anchorage-independent growth in TNBC [66]. This indicates that the role of autophagy cannot simply be referred to the subtype, but the stage of the disease and its autophagic properties need to be addressed as well. Thus, it is important to shed further light on the role of autophagy in breast cancer, especially concerning potential prognostic and predictive autophagy markers in human patients, since most studies have been conducted in vitro or in mouse models. A focus on developing and clinically testing more specific autophagy modulators is vital to provide a therapeutic benefit for patients. Given the heterogeneous nature of breast neoplasms and the fact that cancer treatment may require long-term administration of certain autophagy modulators, there is a great need to foster research on biomarkers to monitor autophagy status in vivo upon novel combination treatments. It is important to keep in mind that autophagy can modify tumor immune responses as well [207,208,209,210]. Altogether, identifying and validating biomarkers for active autophagy and the use of specific autophagy modulating drugs might enable researchers and medical oncologists to predict the effects of early versus late autophagy modulation for personalized breast cancer care.

## Figures and Tables

**Figure 1 cells-10-01447-f001:**
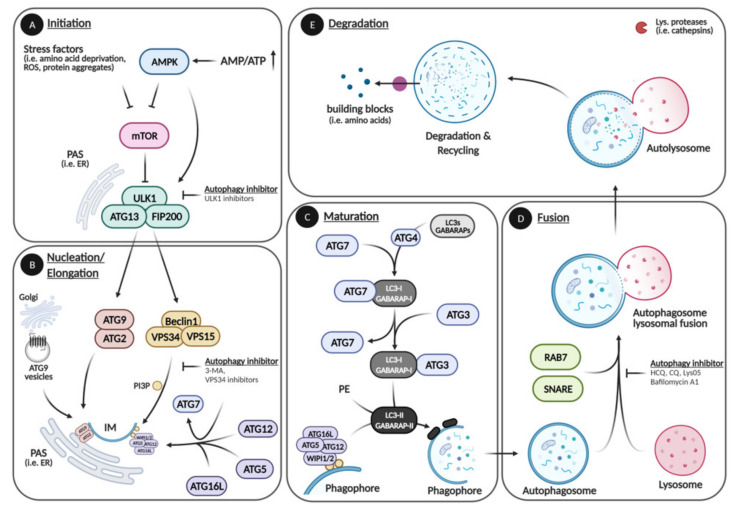
Autophagic pathway. The autophagic pathway can be divided in five main steps: Initiation, Elongation, Maturation, Fusion, and Degradation. (**A**) During initiation, stress factors (i.e., low energy, amino acid deprivation, reactive oxygen species (ROS), protein aggregates) activate ULK1 directly or indirectly by inhibiting mTOR. (**B**) The initiator kinase ULK1 recruits and activates the ATG9–ATG2 complex and the Beclin1 complex that initiate phagophore formation at the PAS. The Beclin1 complex further facilitates membrane elongation by recruiting the first ubiquitin-like conjugation system composed of ATG5, -12, and -16L. (**C**) During maturation, the second ubiquitin-like conjugation system that includes LC3s (most notably LC3B) and other ATG8 homologs is formed. The ATG8 homologs are cleaved (LC3-I, GABARAP-I) by the ATG4 protease followed by the conjugation with phosphatidylethanolamine (PE) (LC3-II, GABARAP-II) and incorporation into the isolation membrane (IM) via ATG5/12 /16L and ATG3/7 complexes, which leads to the elongation and closure of the autophagosome, thereby incorporating cytoplasmic content. (**D**,**E**) When in close proximity, autophagosomes and lysosomes fuse, and the cargo is degraded into cellular building blocks (i.e., amino acids) by lysosomal proteases (cathepsins) via hydrolysis. ULK1 inhibitor: Autophagy-activating kinase 1 inhibitor. 3-MA: 3-Methyladenine. VPS34 inhibitor: PI3K inhibitors. HCQ (Hydroxychloroquine), CQ (Chloroquine), Lys05: Lysosomal lumen alkalizers. Bafilomycin A1 (Baf. A1): Vacuolar type H-ATPase inhibitor. Please see text for further information. Created with Biorender.com, accessed on 3 May 2021.

**Figure 2 cells-10-01447-f002:**
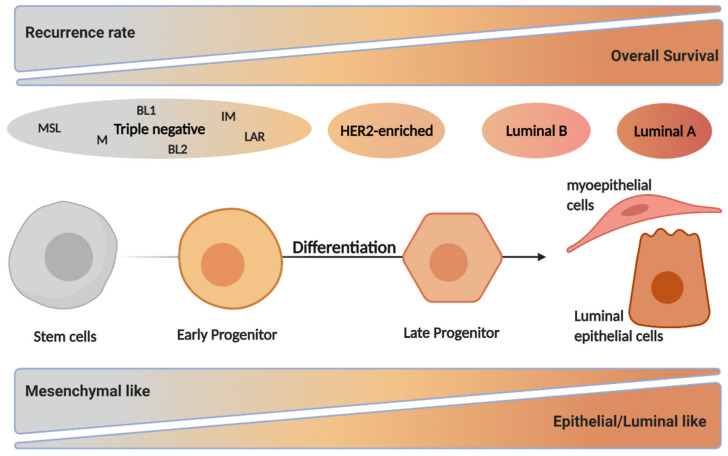
The different intrinsic breast cancer subtypes in relation to mammary differentiation. Genomic studies identified four different breast cancer subtypes: luminal A, luminal B, HER2-enriched, and triple-negative. These subtypes differ in the expression profile of hormonal receptors including ER, PR, and HER2 and in proliferation markers such as Ki-67. The overall survival is highest in luminal A, while triple negative have the highest recurrence rate, leading to the worst prognosis. The subtypes also differ in their phenotype since triple-negative cancers show basal- and mesenchymal-like features, while luminal A breast cancers have an epithelial- and luminal-like phenotype. Of note, the breast cancer subtypes share common features with cells at distinct steps of the mammary differentiation, representing the differentiation hierarchy of healthy mammary cells in which the initial oncogenic insult occurred. Created with BioRender.com, accessed on 3 May 2021.

**Figure 3 cells-10-01447-f003:**
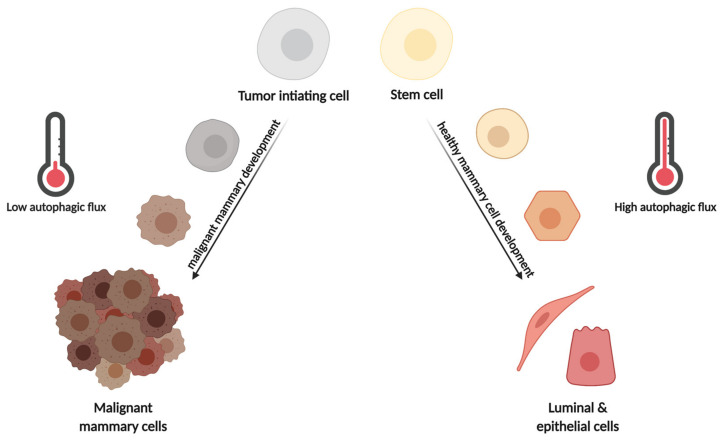
Effects of autophagy during normal and malignant mammary development. During healthy mammary cell differentiation, high levels of autophagy are needed to protect the cells from various insults such as metabolic and ROS-induced stress. On the other hand, the development of malignant tumors from tumor-initiating cells (TIC) requires low levels of autophagy, since high levels of autophagy maintain TIC dormancy by protecting from metabolic stress. In parallel, low autophagic activity stabilizes Pfkfb3 that promotes cell cycle progression and blocks apoptosis. Thus, high autophagic activity is required in benign mammary development, while low activity is necessary for malignant mammary development. Created with BioRender.com, accessed on 3 May 2021.

**Figure 4 cells-10-01447-f004:**
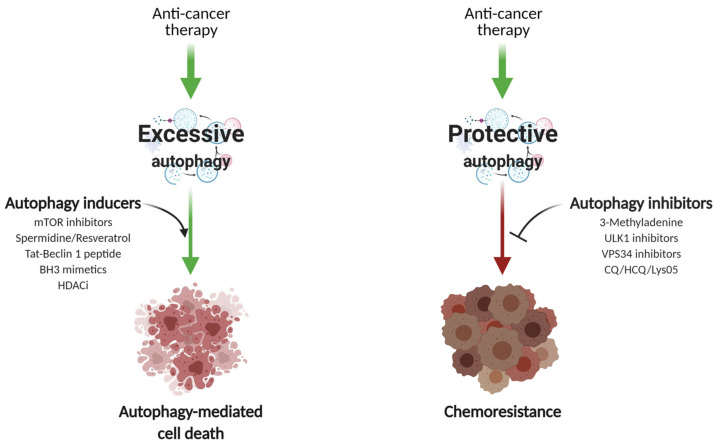
Therapeutic modulators of autophagy. The effect of autophagy modulation on the therapeutic benefit of breast cancer largely depends on the stage of disease and prior treatment course. Autophagy is frequently upregulated in tumor cells as a stress response to a given anti-cancer therapy (upper part). Commonly applied chemotherapeutic drugs may induce excessive or sustainable autophagy, mediating so-called autophagic cell death in the cancer cells (left path). In this case, agents capable of increasing autophagic flux (such as novel mTORC1/2 kinases, Bcl-2 or HDAC-selective inhibitors, etc.) may be applied to enhance the cytotoxic effect. In contrast, enforced autophagy upon chemotherapy may result in cytoprotective effects (right path) and thereby trigger drug resistance, leading to tumor relapse. Therefore, a range of inhibitory compounds that aimed to disrupt the autophagic machinery at different stages can sensitize cancer cells to the proposed therapy. Created with BioRender, accessed on 3 May 2021.

**Figure 5 cells-10-01447-f005:**
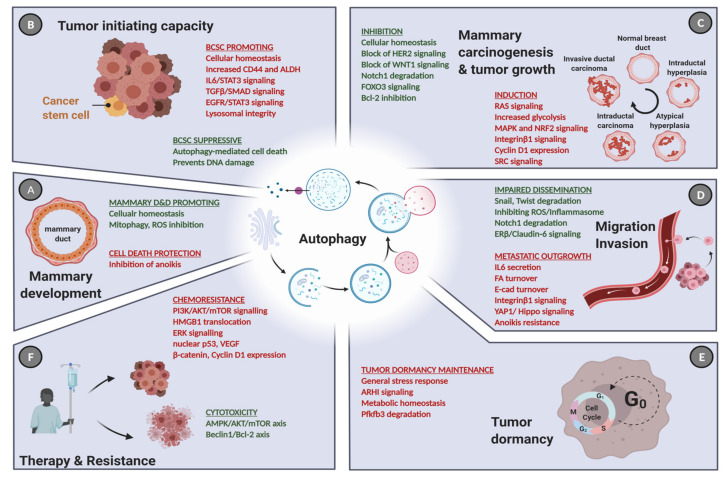
The role of autophagy in cancer initiation, progression, and treatment. (**A**) Apart from its ambiguous role in normal mammary development and differentiation (D&D), autophagy influences virtually each step of cancer progression in a context-dependent manner. (**B**) Under normal conditions, high autophagic activity is predominantly cytoprotective and hence prevents cancer initiation. At the same time, autophagy facilitates the maintenance of CD44^+^/ALDH^+^ BCSC population. (**C**) During carcinogenesis, autophagy is frequently hijacked to support tumor expansion by promoting activation of the Ras pathway and the NRF2 oncogene, particularly in TNBC. However, inhibitory effects are often seen, which are here exemplified by the contribution of autophagy to block HER2- and WNT1-driven mammary tumorigenesis. (**D**) It is also evident that high autophagy activity may limit EMT traits, since several EMT-transcription factors are degraded via autophagy, including SNAIL and TWIST. In contrast, the inhibition of anoikis in malignant cells and accelerated cell migration (due to turnover of focal adhesions) were also attributed to enforced autophagy. (**E**) Furthermore, high levels of autophagy maintain the dormancy of the BCSCs, which is a main source of disease relapse, protecting them from metabolic stress. (**F**) Commonly induced autophagic activity upon anti-cancer therapy leads to an excessive autophagy potentiating drug cytotoxicity via the AMPK/mTOR/ULK pathway. In contrast, increased autophagy activates ERK and p53 signaling to limit the effect of chemotherapeutics, facilitating drug resistance. Green: tumor-suppressive roles, red: oncogenic roles. Created with BioRender.com, accessed on 3 May 2021.

## Data Availability

Not applicable.

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
