# Peer review of "The Multifaceted Functions of Autophagy in Breast Cancer Development and Treatment"

_cells, 2021, doi:10.3390/cells10061447_

Round 1

Reviewer 1 Report

This manuscript reviewed recent advances in the function of autophagy in breast cancer development and treatment. The author summarized current knowledge on autophagy, and then explained how autophagy research have implicated in tumorigenesis and tumor progression, cancer stem cells, tumor dormancy, and metastasis, in breast cancer. Finally, they described on therapeutic strategies that involve autophagy. I think this manuscript is well written covering several aspects on autophagy along different contexts of the tumor cells. Thus, it would be worth publishing in this Journal. However, there are some concerns listed below that should be considered for revision. Especially, introduction part of autophagy needs to be updated.

              If possible I would like authors to consider that autophagy does not necessarily include whole lysosomal function. Thus, lysosome inhibitors also suppress degradation of materials that are incorporated along endocytic pathway. Similarly, lysosome malfunction should cause autophagy suppression even under conditions where autophagy initiation is upregulated. Also, previous reports have proposed non-autophagic functions for some ATGs and related factors. I think these considerations would ease complexity of autophagy in cancer development and treatment.

  1. In the first section of “Autophagy” some words or explanations are lacking as follows,

- In Fig. 1, amino acid deprivation should be included as an induction signal.

- Atg2 and Atg9 (vesicles) should be briefly included in the text, and in the elongation step in Fig. 1, because recent papers propose lipid transfer functions for these molecules.

- The Golgi-like images as the PAS in Fig. 1 may mislead readers. I understand that the Golgi membrane could be one of sources of phagophore membrane, but it is considered as a minor event and there have been no morphological evidence that show direct transformation from the Golgi cisterna into phagophore.

- “Autophagophore” should be just “Phagophore” in Fig. 1.

- I think “LC3” should be mentioned in this section.

- there is no explanation for “Atg8-I” and “Atg8-II” in the Fig. 1.

- Relationship between WIPI-1, Atg5-12-16L complex, Atg3, and Atg7 (lacking) in Fig. 1 is somewhat different from the corresponding main text.

- According to recent papers, Atg8 is not required for the elongation step, since phaogophore can be formed in the absence of Atg8s.

  1. Other minor comments are listed below

Line 88: In line “with”?

Line 164: “…at late degradation stages…” might be better.

Line 182: There is no description on stem cells in previous sentences.

Line 187: The citation [49] on Beclin 1 appears wrong.

Line 487: Is the T47D cells derived from breast cancer?

Reviewer 2 Report

Dear Author,

The review article titled "The multifaceted functions of autophagy in breast cancer development and treatment" by Nicolas et al.,  summarized well.

I have few minor comments related to the review article.

Minor Changes:

  1. Fig 1 autophagy pathway needs few updates. LC3B is one of the markers for autophagy induction and it involved in the autophagosome formation. This must be mentioned in the figure.
  2. Similarly, the author can mention lysosomal enzymes in the degradation pathway and finally, the cargo degraded into amino acids (blue dots),.
  3. The author can mention where the autophagy inhibitors are involved in the pathway.
  4. What is the role of autophagy inhibitors(3-MA, Baf-1, and CQ) in breast cancer treatment?
  5. what is the mechanism behind cytotoxic autophagy in breast cancer treatment?

Reviewer 3 Report

Manuscript written by Niklaus et al., as a review paper, describes a role in development and treatment of breast cancer specifically. This review focuses on the mechanisms of autophagy during breast carcinogenesis specifically and discusses the role of autophagy in the traits of aggressive breast cancer cells such as migration, invasion and therapeutic resistance. By dividing text in several chapters the authors inform readers about autophagy process, breast cancer, the role of autophagy in development of normal breast and breast cancer, then breast stem cell, breast cancer dormancy, metastasis and therapy. Some of these topics are already described in the literature but through this review they are updated. The authors successfully emphasized the current role of autophagy in different aspects of breast cancer pathology and treatment. They further discussed, based on literature data, that the autophagy in breast cancer is versatile and dependents on the biology of the breast cancer subtype, stage of the disease and its autophagic properties. The literature is up to date, it is well written and easy to follow due to included illustrations. I am recommending this manuscript for publication in Cells.
Minor
1) Remove bold from the title of Figure 5 since the rest of titles are not written in bold

Author Response

We changed the title to standard font.